# Finding the Right Way to Target EGFR in Glioblastomas; Lessons from Lung Adenocarcinomas

**DOI:** 10.3390/cancers10120489

**Published:** 2018-12-04

**Authors:** Ya Gao, Wies R. Vallentgoed, Pim J. French

**Affiliations:** Department of Neurology, Erasmus MC Cancer Institute; 3015 CD Rotterdam, The Netherlands; y.gao@erasmusmc.nl (Y.G.); w.vallentgoed@erasmusmc.nl (W.R.V.)

**Keywords:** EGFR, glioblastoma, glioma, pulmonary adenocarcinoma, lung cancer, erlotinib, gefitinib, lapatinib

## Abstract

The *EGFR* gene is one of the most frequently mutated and/or amplified gene both in lung adenocarcinomas (LUAD) and in glioblastomas (GBMs). Although both tumor types depend on the mutation for growth, clinical benefit of EGFR tyrosine kinase inhibitors (TKIs) has only been observed in LUAD patients and, thus-far, not in GBM patients. Also in LUAD patients however, responses are restricted to specific *EGFR* mutations only and these ‘TKI-sensitive’ mutations hardly occur in GBMs. This argues for mutation-specific (as opposed to tumor-type specific) responses to EGFR-TKIs. We here discuss potential reasons for the differences in mutation spectrum and highlight recent evidence for specific functions of different *EGFR* mutations. These mutation-specific effects likely underlie the differential treatment response between LUAD and GBMs and provide new insights into how to target EGFR in GBM patients.

## 1. Introduction

More than fifty years ago, Stanley Cohen reported on the isolation of a heat-stable protein from mouse salivary glands that was able to induce the eruption of incisors and separation of the eyelids [1]. Later studies showed that the eyelid separation was actually a consequence of enhanced keratinization and growth of the epidermis and, based on this function, the protein was coined epidermal growth factor (EGF) [2,3]. Around a decade later, the same group found evidence for a receptor for the growth factor on human fibroblasts and another decade later, the receptor was cloned and sequenced [4,5].

The *EGF* receptor (*EGFR*) gene encodes for a protein of 1210 amino acids which is built up of an extracellular ligand binding domain, a single transmembrane domain, a juxtamembrane region, an intracellular tyrosine kinase domain and a C-terminal regulatory domain. The extracellular region can be divided into four subdomains, domains I–IV, that contain two leucine rich regions (I and III, also known as L1 and L2) and two cysteine rich domains (II and IV, also known as CR1 and CR2). Proteins with a similar domain structure include ERBB2, ERBB3 and ERBB4 (which are also known as HER2-4), and are collectively known as the ERBB protein family, which are all members of the receptor tyrosine kinase superfamily.

In the absence of ligand, EGFR exists as a monomer (or as an inactive dimer) on the plasma membrane. When ligands are available, they bind to the extracellular leucine rich domains I and III, and this association results in the exposure of the cysteine rich domain II, which allows for receptor dimerization. The dimerization then induces phosphorylation of tyrosine residues on the C-terminal intracellular domain of the dimerization partner (trans-phosphorylation) [6]. Phosphorylation occurs on multiple sites: the intracellular domain contains twenty tyrosine residues, twelve of which can be phosphorylated following ligand binding [7]. This phosphorylation results in the recruitment of specific adaptor proteins and subsequent activation of signal transduction cascades. These cascades include the RAS-RAF-MEK-ERK, PI3kinase-AKT-mTOR, SRC, JNK, PLC-γ-PKC and STAT pathways and ultimately result in cellular proliferation, migration and survival [8,9].

Although the EGF receptor (EGFR) was identified based on its affinity for EGF, later studies showed it can be also activated by six other, related- ligands: heparin-binding EGF-like growth factor (HB-EGF), amphiregulin, epiregulin, epigen, betacellulin and transforming growth factor-alpha (TGF-α) [10]. EGF, TGF-α, HB-EGF and betacellulin are the ligands that bind with high affinity to EGFR; amphiregulin, epiregulin and epigen bind with 10–100 fold less affinity. Each ligand can induce quantitative differences in responses, but can also elicit ligand-specific responses [10,11].

## 2. *EGFR* Mutations in LUAD and Gliomas

Cancer is caused by the accumulation of acquired genetic changes in oncogenes and tumor suppressor genes. The epidermal growth factor receptor (*EGFR*) gene is a key oncogene that is mutated in many tumors including lung adenocarcinomas (LUAD) and glioblastomas (GBM). In LUAD, around 90% of all *EGFR* mutations comprise of either short in-frame deletions in exon 19 (in particular around residues 747–750) or the *L858R* missense mutation in exon 21 [12,13]. Other, less common mutations in LUAD include *G719X* missense mutations (~3%) and in-frame insertions in exon 20 (~3%) [14]. All these mutations result in increased and continuous EGFR phosphorylation and activation [15,16].

In GBMs, the initial driving event is thought to be high copy number amplification of the *EGFR* gene, present in tumor cells as double minutes (extrachromosomal circular DNA fragments) with levels ranging from >5 to more than 100 copies per cell [17]. These double minutes likely increase the copy number (and RNA expression) of the oncogene more effectively compared to chromosomal amplification. Double minutes are unevenly distributed across the two daughter cells during cell division which enhances tumor heterogeneity and plasticity [17]. Amplification of the *EGFR* gene is followed by the acquisition of a plethora of mutations that include intragenic deletions, point mutations and gene-fusions [18]. Multiple *EGFR* mutations may be present within the same tumor which also contributes to tumor heterogeneity [19]. The most common *EGFR* mutation in GBM is the in-frame deletion of exon 2–7, coined *EGFRvIII* and occurs in ~50% of all *EGFR*-amplified GBM cases. EGFRvIII has impaired ligand binding and is constitutively active, though its activity is only ~10% of endogenous EGFR signaling [20]. Missense mutations that are commonly found in GBM are often located on the extracellular domain and include A289X, G598X and R108K mutations. These mutations also result in a constitutively active protein and increase the tumorigenic potential of cells [19].

Both in lung cancer and in GBMs, *EGFR* mutations are driver mutations and the tumors remain dependent on this oncogene for growth [21,22,23,24]. Despite the similarities in activity, the most prominent mutations in LUAD, exon-19 deletions and L858R, have, to date, never been identified in GBMs and the most common mutation in GBMs, EGFRvIII, has never been identified in LUAD. This indicates that each tumor type has an almost (see below), unique mutation spectrum [14,20].

## 3. Clinical Activity of EGFR-TKIs in LUAD, But Not GBM Patients

It is well known that EGFR-TKIs provide remarkable survival benefit to patients with *EGFR*-mutated LUAD. These benefits were initially discovered by research on clinical trials that examined the efficacy of EGFR-TKIs in LUAD patients, in which antitumor activity and an increase in survival was observed in patients who failed on prior chemotherapy [25,26]: translational research showed that the clinical responses were correlated to mutations in *EGFR* [27,28]. Landmark studies such as the IPASS study showed that gefitinib improved the 12 months progression free survival in advanced, previously untreated pulmonary adenocarcinoma patients, but only in those patients where activating mutations in the *EGFR* gene were identified. Similar improvements were observed in *EGFR*-mutated, metastatic non-small cell lung cancer patients [29,30]. A phase III study that included only *EGFR*-mutated lung cancer patients confirmed these observations [29].

Because of the clinical responses observed in *EGFR*-mutated LUAD patients, and because of the high frequency of *EGFR* mutations in GBMs, it was logical to test the clinical efficacy of EGFR-TKIs in GBM patients. Although several such trials have been conducted (Table 1), thus-far none of these demonstrated a clear clinical benefit of the inhibitors, despite inhibitors showing target inhibition on the various EGFR mutations in preclinical models [19]. For example, two studies conducted in primary gliomas showed no additional clinical benefit of adding gefitinib after radiotherapy [31,32]. Similar disappointing data were obtained in two studies on recurrent gliomas where single agent erlotinib did not improve the 6 months progression-free survival [33,34].

There are several possibilities why those trials failed to improve on their primary endpoint. First of all, the trials may not have included the right patient population. While this seems trivial, trials testing EGFR-TKIs selected patients based on histological criteria and did not select for *EGFR*-mutated tumors [31,33,43,45,50]. However, the frequency of *EGFR* amplification and mutations are sufficiently high in GBM patients (estimated to be ~50% of all GBMs) to show some signal of efficacy. Moreover, translational research on the material from these trials also failed to show clinical improvement in the samples that had *EGFR*-mutations [31,34,51,52]. A second possibility for trial failure is that the concentration of drug does not reach sufficiently high concentrations in the tumor, for example by lack of penetration through the blood brain barrier. Indeed, intratumoral drug concentrations for erlotinib were much lower of that in plasma and erlotinib did not affect intratumoral EGFR signaling [33]. In contrast, a phase II trial in which the intratumoral levels of gefitinib was measured in 22 patients showed concentrations sufficiently high to inhibit the phosphorylation of EGFR [43]. 

A third option for therapy refractoriness of GBMs to EGFR-TKIs is that these tumors no longer depend on the oncogene for growth. However, biological experiments conducted in mice and using primary patient cell lines showed that these tumors do remain dependent on EGFR and therefore do not explain why GBMs do not respond to EGFR-TKIs [21,22]. It is possible that GBMs have an innate resistance to EGFR-TKIs, such as the upregulation (or coactivation) of PDGFRA and cMET [53]. Such an innate resistance has been identified in colon carcinomas where inhibition of one tyrosine kinase (BRAF) is bypassed by the activation of another (EGFR) [54]. Alternatively, perturbation of downstream pathways such as PTEN deletion may also confer resistance to EGFR TKIs [40]. It remains however to be determined if, and if so which-, changes (genetic or epi-genetic) underlie the therapy refractoriness of GBMs.

The dynamics of *EGFR* mutations can also play a role in treatment resistance of GBMs. Since *EGFR* is present as double minutes, cell division can result in the asymmetric distribution of *EGFR* copynumber and variants, and thus in rapid selection of potentially resistant clones [17]. An example of dynamics is the expression of *EGFRvIII* which can be restricted to certain regions of the tumor only and can change over time [55,56,57]. Our unpublished observations show that temporal dynamics is not restricted to *EGFRvIII* but also true for other *EGFR* mutations. If only one of those mutations is resistant to the TKI, the dynamics of double minutes likely results in a rapidly acquired resistance to EGFR-TKIs.

## 4. Different Mutations Activate Different Pathways and May Explain Refractoriness to EGFR-TKIs

On the functional level, several lines of evidence suggest that different mutations in *EGFR* activate unique signal transduction pathways. For example, EGFRvIII and EGFRwt have differential activation of the JNK, STAT and MAPK signaling pathways and induce the expression of a unique set of genes [58,59,60,61,62]. We have also demonstrated that EGFRwt, EGFRL858R and EGFRvIII each bind to a unique set of proteins and activate different molecular pathways [63]. If mutation-specific pathways active in GBMs are independent of EGFR phosphorylation, they explain why GBMs do not respond to EGFR-TKIs. For example, EGFR contains a functional nuclear localization signal and has been found present at high levels in the nucleus where it associates with the promoter of cell proliferation genes [64,65]. However, the nuclear accumulation is mutation dependent: mutations found in GBMs (e.g., EGFRvIII) accumulate in the nucleus whereas LUAD-specific mutations such as EGFR L858R do not [66,67,68]. Other studies have shown that the nuclear accumulation is independent of EGFR inhibitors [65,69]. It is therefore possible that activation of mutation-specific and TKI-independent pathways contribute to the lack of response to EGFR-TKIs. Which pathways are differentially activated and whether (co-) targeting these pathways will ultimately benefit GBM patients remains to be determined.

However, one argument against activation of mutation-specific pathway activation is the observation that some mutations have been found in both tumor types. Yes, the type of mutations vastly differs between LUAD and GBMs: exon 19 deletions or the *EGFR L858R* point mutations have never been found in GBMs and *EGFRvIII* is not present in LUAD [18,20]. Sporadically however, exon 20 mutations, though rare in LUAD, have also been found in GBM patients [18]. Indeed, in a recent sequencing effort of ~200 GBM samples performed by our group, we identified two patients with mutations near identical to those found in LUAD (p.H773_V774insAH and p.H773dup) (Figure 1). Conversely, sporadic R108K, A289A/T and G598V mutations that are common in GBMs have been identified in LUAD (Figure 1) [70,71,72]. This overlap in mutational spectrum, though rare, may suggest that LUAD and GBMs are unlikely to activate mutation-specific pathways. However, since multiple mutations accumulate in the *EGFR* gene in GBMs, other activating mutations may confer potential mutation-specific effects.

## 5. Lessons from LUAD: Not all LUAD Respond to EGFR-TKIs and Not All TKIs are Effective in LUAD

Mutations that show response to EGFR-TKIs in LUAD have sporadically been identified in other tumor types. In various tumors harboring ‘classical’ exon 19 deletions or the L858R point mutation, EGFR-TKIs have shown clinical responses [73,74,75]. Such responses have also been documented for the less common, but responsive, G719X mutations, also in different tumor types [76,77]. The fact that sensitive mutations in LUAD appear also sensitive to TKIs in other tumor types indicates that response to EGFR-TKIs is not specific to the type of tumor, but is specific for the mutation present. This can have important consequences for GBM patients, as sporadic G719X mutations have been identified (Figure 1) [72,78].

This mutation-specific response is also conversely observed in LUAD patients: while the most common mutations in LUAD are sensitive to EGFR-TKIs, LUAD patients harboring exon 20 mutations are largely insensitive [79,80]. This lack of responsiveness was also observed with these mutations in various preclinical model systems [81,82]. Although rare, they comprise of ~3% of all *EGFR* mutations in LUAD and are considered to be activating mutations. Interestingly, and despite the lack of a clinical response, addition of TKIs does result in EGFR dephosphorylation in preclinical model systems [83,84]. The observation that not all *EGFR*-mutated LUAD respond to EGFR-TKIs indicate that response is limited to a defined set of mutations only.

Apart from the mutation-specificity of the response, there also is evidence for drug-specificity: although several EGFR-TKIs (erlotinib, gefitinib, afatinib, dacomitinib and osimertinib) have provided clinical benefit to *EGFR*-mutated LUAD patients, a phase II study on lapatinib did not show any signs of clinical activity [85]. This lack of clinical activity is surprising as lapatinib, similar to the other listed TKIs, is a highly potent inhibitor of EGFR phosphorylation. Erlotinib and gefitinib were the first to be developed and are both reversible inhibitors. Afatinib and dacomitinib are second generation irreversible inhibitors and osimertinib is a third generation inhibitor that also inhibits EGFR containing the T790M resistance mutation. Although all inhibitors inhibit EGFR phosphorylation, each drug has its own unique binding pocket, kinetics and properties (see e.g., [86]). Of note in this is that drugs like erlotinib associate with the active conformation while lapatinib traps the protein in an inactive conformation [87,88,89]. Perhaps this drug-specific conformational association underlies the marked difference in treatment response.

Nevertheless, the lack of clinical activity to lapatinib indicates that response to EGFR-TKIs is specific to the drug that was used. If so, response to inhibitors is a highly specific event, in which only specific mutations (exon 19 deletions, L858R and G719X missense mutations), and in the context of specific drugs only provide clinical benefit. The absence of either will result in clinical inactivity. Importantly, this would mean a shift in research paradigms from understanding why certain tumors do not respond to research into why only specific mutations do respond. Understanding why tumors respond will then lead to better design of novel targeted treatments.

## 6. Conclusions

Only specific mutations in EGFR respond to treatment, and this response seems independent of the type of tumor. Conversely, activating mutations with lack of clinical response to EGFR-TKIs, such as those found in GBMs, also do not respond to treatment in LUAD patients. Treatment response therefore is not tumor-type dependent, but mutation dependent. Moreover, only specific drugs seem to provide clinical benefit, despite showing similar target inhibition. Response to inhibitors therefore appears to be a highly specific event: only specific mutations in the context of specific drugs provide clinical benefit. The absence of either will result in clinical inactivity. The specificity of response warrants further investigation into its mechanisms as understanding why tumors respond will lead to better design of novel targeted treatments.

## Figures and Tables

**Figure 1 cancers-10-00489-f001:**
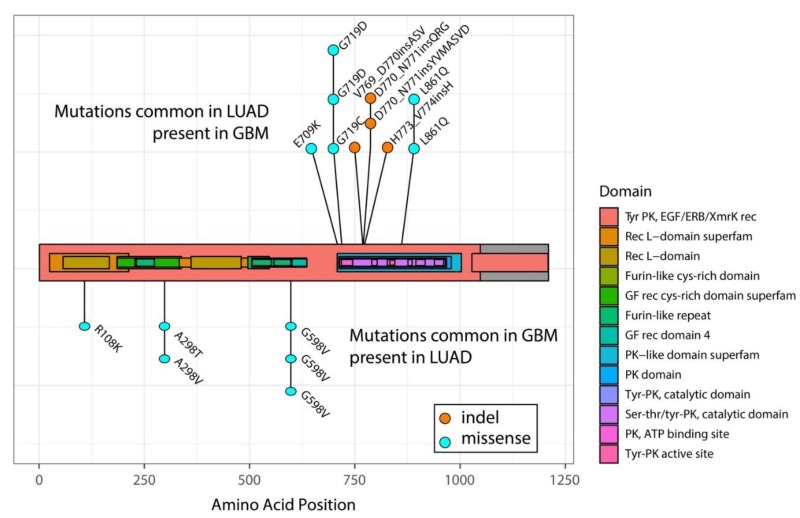
Mutations common to LUAD are sporadically identified in GBMs and vice versa. Each dot reflects a single sample identified with the specific mutation. Domains of EGFR are highlighted in color. PK: protein kinase; GF: growth factor; rec: receptor. The transmembrane is localized around amino acids 622–644.

**Table 1 cancers-10-00489-t001:** clinical trials of EGFR-TKIs in gliomas.

Drug	Phase	Clinical Trial ID	Comparator	Histology	*n*	Ref.
Erl + TMZ/RT	I/II	NCT00039494	single arm	GBM	97	[35]
Erl + Bev	II	NCT00671970	single arm	GBM, AG	57	[36]
Erl + Soraf	II	NCT00445588	single arm	rGBM	56	[37]
Erl + Bev + TMZ	II	NCT00525525	single arm	GBM, GSC	74	[38]
Erl + TMZ	II	NCT00187486	single arm	GBM, GLS	65	[39]
Erl	I/II	NCT00301418	single arm	GBM, AA	11	[40]
Erl + Sirol	II	NCT00672243	single arm	rGBM	32	[41]
Erl + Sirol	I/II	NCT00112736	single arm	rGlioma	69	[42]
Erl	II	NCT00250887	TMZ/BCNU	rGBM	110	[34]
Gef	II	NCT00250887	single arm	rGBM	22	[43]
Gef + Cedir	II	NCT01310855	Cedir	rGBM	38	[44]
Gef	II	NCT00016991	Single arm	rGBM	57	[45]
Lap	I/II	NCT00099060	single arm	rGBM	17	[46]
Afa	I/II	NCT00727506	TMZ/afa, TMZ	rGBM	119	[47]
Dac	II	NCT01520870	single arm	rGBM	49	[48]
Sun	II	NCT00923117	Bev naïve/resistant	GBM	72	[49]

rGBM: recurrent or progressive GBM; AG: anaplastic glioma, rGlioma: recurrent glioma; GLS: gliosarcoma; AA: anaplastic astrocytoma. Erl: erlotinib; TMZ: temozolomide; RT: radiotherapy; soraf: sorafenib; sirol: temsirolimus; cedir: cediranib; bev: bevacizumab; lap: lapatinib; afa: afatinib; Dac: dacomitinib; Sun: sunitinib.

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
