# Peer review of "Finding the Right Way to Target EGFR in Glioblastomas; Lessons from Lung Adenocarcinomas"

_cancers, 2018, doi:10.3390/cancers10120489_

Reviewer 1 Report

I read the short manuscript by Gao and Colleagues in a very short time: it presents plainly and clearly an interesting observation, on the responses of glioblastoma and lung adenocarcinoma patients to drugs targeting EGFR. The conclusion suggests a paradigm shift in approaching the relationship drug-type of cancer, by highlighting the similarities in responses to drugs in different tumors which harbor similar mutations. This is an interesting point that may advance our understanding of cancer responses and deserves publication.

Author Response

We would like to thank the reviewer for critically reading our manuscript.

Reviewer 2 Report

Gao et al. herein discuss potential reasons for the differences in mutation spectrum in cancer and highlight recent evidences for specific functions of different EGFR mutations, focusing on two major neoplasms with EGFR gene frequently mutated and/or amplified including lung adenocarcinoma (LUAD) and glioblastoma (GBM). The manuscript is concise and well written to explain differential treatment response between LUAD and GBMs and provide new insights into how to target EGFR in GBM patients, but several minor modifications are necessary to further clarify their arguments.

1. In Figure 1, each domain is relatively hard to see, which should be revised. Also, it would be preferable if they depict the schematic cell membrane with EGFR inserted.

2. Looking at Figure 1, it seems that mutations common in GBM tend to occur in the extracellular domain whereas mutations in LUAD in the kinase domain. Does this difference possibly affect the therapeutic efficacy with TKIs?

3. As a GBM’s resistance mechanism, background genotype (e.g. PTEN mutation) and rewiring of intracellular signaling circuits are also important, and they should add these in the text by citing references.

4. In relation to comment 3, it would be better to make the scheme as a new Figure to display the mechanisms why GBM demonstrate the resistance to TKIs.

5. They provided several examples of TKIs (erlotinib, gefitinib, afatinib, dacomitinib and osimertinib), and they should depict the characteristics of each inhibitors, which will make it easier for the readers to understand why each drug hit the specific target or not.

6. Please make one more table to show the effective combination of the specific mutation and specific TKI in relation to the comment 5.

7. They should provide one more section to demonstrate the actual and theoretical attempt to overcome this mutation-specific resistance to TKIs in GBM.

8. Is there any other mutation in cancer which exhibits this type of mutation-specific resistance to treatments? If so, please provide a few sentences to introduce other examples to generalize their theory.

9. Several abbreviated words are not spelled out in the text. Please check through the manuscript.

10. Please fix several minor English grammatical errors.

Author Response

1. In Figure 1, each domain is relatively hard to see, which should be revised. Also, it would be preferable if they depict the schematic cell membrane with EGFR inserted.

We have modified the figure so that the domains are better visible, and have included the legend for the individual domains. We have also stated the location of the transmembrane domain.

2. Looking at Figure 1, it seems that mutations common in GBM tend to occur in the extracellular domain whereas mutations in LUAD in the kinase domain. Does this difference possibly affect the therapeutic efficacy with TKIs?

Mutations in GBMs tend to occur in the extracellular domain and they are considered activating (Lee et al, Plos medicine 2006). However, their activity can be effectively inhibited by EGFR TKIs, and this inhibition therefore does not explain therapy refractoriness of GBMs. To address this question by the referee, we have incorporated the statement: “.... inhibitors showing target inhibition on the various EGFR mutations in preclinical models” in our manuscript.

  3. As a GBM’s resistance mechanism, background genotype (e.g. PTEN mutation) and rewiring of intracellular signaling circuits are also important, and they should add these in the text by citing references.

Indeed, it has been reported that only patients expressing EGFRvIII and PTEN show response to EGFR TKIs. We have now incorporated this reference in our manuscript and included the following sentence: “Alternatively, perturbation of downstream pathways such as PTEN deletion may also confer resistance to EGFR TKIs” (line 135).

4. In relation to comment 3, it would be better to make the scheme as a new Figure to display the mechanisms why GBM demonstrate the resistance to TKIs.

Our manuscript highlights several aspects why GBM patients do not respond to TKIs: incorrect patient selection, insufficient intratumoral drug concentrations, independency of tumor growth on the receptor, innate resistance, receptor dynamics, mutation-specific pathway activation and mutation specific responses to TKIs. We feel it would be very difficult to envision all these options in a single figure.

5. They provided several examples of TKIs (erlotinib, gefitinib, afatinib, dacomitinib and osimertinib), and they should depict the characteristics of each inhibitors, which will make it easier for the readers to understand why each drug hit the specific target or not.

I’m not entirely sure what the reviewer means with the characteristics of each inhibitor. They have all been developed to inhibit EGFR phosphorylation and feel that an in-depth analysis of the differences between each inhibitor would be far beyond the scope of this manuscript. We have however briefly summarized some differences in the manuscript to address this comment of the referee: ‘It should be noted that all inhibitors have been developed to inhibit EGFR phosphorylation. Erlotinib and gefitinib were the first to be developed and are both reversible inhibitors. Afatinib and dacomitinib are second generation irreversible inhibitors and osimertinib is a third generation inhibitor that also inhibits EGFR containing the T790M resistance mutation.

6. Please make one more table to show the effective combination of the specific mutation and specific TKI in relation to the comment 5.

No database lists individual mutations with individual drugs. Even the most comprehensive databases limit themselves to lumping several mutations (e.g. exon 19 deletions) and drugs together. The reason is simply because there too many different mutations and too many drugs to make such a table, and for many of the rarer mutations response to TKIs is either not known or anecdotal and therefore has to be inferred by similarity. However, we do agree with the referee that listing the most common responsive mutations would help. Accordingly, these have now been mentioned (exon 19 deletions, L868R,  and G719X missense mutations) in the main body of the manuscript.

 7. They should provide one more section to demonstrate the actual and theoretical attempt to overcome this mutation-specific resistance to TKIs in GBM.

This is quite an impossible feat to do:  Even in lung cancer where most exon-20 insertions/deletions do not respond to treatment, even though the TKIs inhibit EGFR phosphorylation, we simply do not know why tumors with such mutations do not respond to EGFR-TKIs.    

8. Is there any other mutation in cancer which exhibits this type of mutation-specific resistance to treatments? If so, please provide a few sentences to introduce other examples to generalize their theory.

I believe this would be beyond the scope of this manuscript. The story around EGFR is already very complex, and including additional drugs and additional mutation-specificity would mean basically a review for each and every drug + mutation combination out there.

9. Several abbreviated words are not spelled out in the text. Please check through the manuscript.

The manuscript has been checked for unexplained abbreviations

10. Please fix several minor English grammatical errors.

The manuscript has been checked for grammatical errors

Reviewer 3 Report

Please proofread once the entire manuscript regarding English grammar, spelling and especially styling preferably by a native English speaker 

Author Response

(The authors gave the same response as above.)

Reviewer 4 Report

Dear Colleagues

You have a great story, I really enjoyed it reading. Despite, I have some thoughts

First, I honestly is not convinced that GBM signaling from "EGFR with mutations" is more valuable then EGFRvIII mediated signaling and therapy resistance. This may be also correct since EGFRvIII may have better surface distribution on glioma surface than EGFR with mutataions.  Secondly, treatment of GBM heterogeneity may result in a raising of GBM clones after therapy. Do you think there is a similar to LUAD tumor distribution of EGFR with mutations in those clones of GBM tissue?

Author Response

First, I honestly is not convinced that GBM signaling from "EGFR with mutations" is more valuable then EGFRvIII mediated signaling and therapy resistance. This may be also correct since EGFRvIII may have better surface distribution on glioma surface than EGFR with mutataions. 

We do not make statements on which mutation is more valuable, we merely highlight the fact that different mutations can have different effects. The nuclear accumulation of EGFR mutations in GBMs actually refers to EGFRvIII, and have specified this in the manuscript.  

Secondly, treatment of GBM heterogeneity may result in a raising of GBM clones after therapy. Do you think there is a similar to LUAD tumor distribution of EGFR with mutations in those clones of GBM tissue?

The presence of sporadic responsive mutations in GBM may indeed result in the appearance of clones resistant to the mutation. I did not address this issue in the manuscript because we firstly need to know whether these tumors actually do respond to the inhibitor. The nature of potentially arising resistant  clones would indeed be very interesting but remains a secondary question and purely speculative.